# Learning Implicit Functions for Topology-Varying Dense 3D Shape Correspondence

**Feng Liu**     **Xiaoming Liu**
Department of Computer Science and Engineering
Michigan State University, East Lansing MI 48824
{liufeng6, liuxm}@msu.edu

## Abstract

The goal of this paper is to learn dense 3D shape correspondence for topology-varying objects in an unsupervised manner. Conventional implicit functions estimate the occupancy of a 3D point given a shape latent code. Instead, our novel implicit function produces a part embedding vector for each 3D point, which is assumed to be similar to its densely corresponded point in another 3D shape of the same object category. Furthermore, we implement dense correspondence through an inverse function mapping from the part embedding to a corresponded 3D point. Both functions are jointly learned with several effective loss functions to realize our assumption, together with the encoder generating the shape latent code. During inference, if a user selects an arbitrary point on the source shape, our algorithm can automatically generate a confidence score indicating whether there is a correspondence on the target shape, as well as the corresponding semantic point if there is one. Such a mechanism inherently benefits man-made objects with different part constitutions. The effectiveness of our approach is demonstrated through unsupervised 3D semantic correspondence and shape segmentation. Code is available at https://github.com/liuf1990/Implicit_Dense_Correspondence.

## 1 Introduction

Finding dense correspondence between 3D shapes is a key algorithmic component in problems such as statistical modeling [4,5,56], cross-shape texture mapping [28], and space-time 4D reconstruction [35]. Dense 3D shape correspondence can be defined as: *given two 3D shapes belonging to the same object category, one can match an arbitrary point on one shape to its semantically equivalent point on another shape if such a correspondence exists.* For instance, given two chairs, the dense correspondence of the middle point on one chair's arm should be the similar middle point on another chair's arm, despite different shapes of arms; or alternatively, declare the non-existence of correspondence if another chair has no arm. Although prior dense correspondence methods [15,18,29–31,37,42,48] have proven to be effective on organic shapes, *e.g.*, human bodies and mammals, they become less suitable for generic topology-varying or man-made objects, *e.g.*, chair or vehicles [22].

It remains a challenge to build dense 3D correspondence for a category with large variations in geometry, structure, and even topology. First of all, the lack of annotations on dense correspondence often leaves *unsupervised learning* the only option. Second, most prior works make an inadequate assumption [50] that there is a similar topological variability between matched shapes. Man-made objects such as chairs shown in Fig. 1 are particularly challenging to tackle, since they often differ not only by geometric deformations, but also by *part constitutions*. In these cases, existing correspondence methods for man-made objects either perform fuzzy [27,47] or part-level [2,44] correspondences, or predict a constant number of semantic points [8,19]. As a result, they cannot determine whether the established correspondence is a "missing match" or not. As shown in Fig. 1(c), for instance, we may

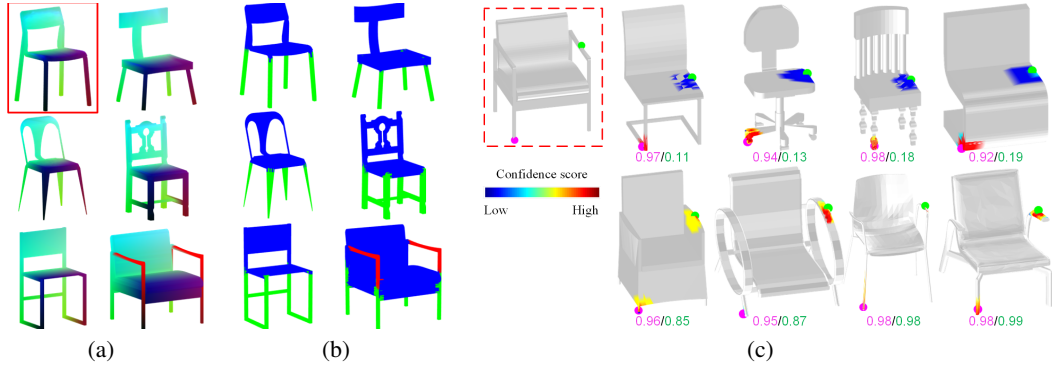

(a)                    (b)                              (c)

Figure 1: Given a set of 3D shapes, our category-specific unsupervised method learns pair-wise dense correspondence (a) between any source and target shape (red box), and shape segmentation (b). Give an *arbitrary* point on the source shape (red box), our method predicts its corresponding point on any target shape, and a score measuring the correspondence confidence (c). For each target, we show the confidence scores of red/green points, and score maps around corresponded points. A score less than a threshold (*e.g.*, 0.2) deems the correspondence as "non-existing"– a desirable property for topology-varying shapes with missing parts, *e.g.*, chair's arm.

find non-convincing correspondences in legs between an office chair and a 4-legged chair, or even no correspondences in arms for some pairs. Ideally, given a query point on the source shape, a dense correspondence method aims to determine whether there exists a correspondence on the target shape, and the corresponding point if there is. This objective lies at the core of this work.

Shape representation is highly relevant to, and can impact, the approach of dense correspondence. Recently, compared to point cloud [1, 40, 41] or mesh [14, 16, 51], deep implicit functions have shown to be highly effective as 3D shape representations [3, 9, 10, 32, 33, 38, 43], since it can handle generic shapes of arbitrary topology, which is favorable as a representation for dense correspondence. Often learned as a MLP, conventional implicit functions input the 3D shape represented by a latent code $\mathbf{z}$ and a query location $\mathbf{x}$ in the 3D space, and estimate its occupancy $O = f(\mathbf{x}, \mathbf{z})$. In this work, we propose to plant the dense correspondence capability into the implicit function by learning a semantic part embedding. Specifically, we first adopt a branched implicit function [9] to learn a part embedding vector (PEV), $\mathbf{o} = f(\mathbf{x}, \mathbf{z})$, where the max-pooling of $\mathbf{o}$ gives the $O$. In this way, each branch is tasked to learn a representation for one universal part of the input shape, and PEV represents the occupancy of the point w.r.t. all the branches/semantic parts. By assuming that PEVs between a pair of corresponding points are similar, we then establish dense correspondence via an inverse function $\hat{\mathbf{x}} = g(\mathbf{o}, \mathbf{z})$ mapping the PEV back to the 3D space. To further satisfy the assumption, we devise an unsupervised learning framework with a joint loss measuring both the occupancy error and shape reconstruction error between $\mathbf{x}$ and $\hat{\mathbf{x}}$. In addition, a cross-reconstruction loss is proposed to enforce part embedding consistency by mapping within a pair of shapes in the collection. During inference, based on the estimated PEVs, we can produce a confidence score to distinguish whether the established correspondence is valid or not. In summary, contributions of this work include:

⋄ We propose a novel paradigm leveraging implicit functions for category-specific unsupervised dense 3D shape correspondence, which is suitable for objects with diverse variations including varying topology.

⋄ We devise several effective loss functions to learn a semantic part embedding, which enables both shape segmentation and dense correspondence. Further, based on the learnt part embedding, our method can estimate a confidence score measuring if the predicted correspondence is valid or not.

⋄ Through extensive experiments, we demonstrate the superiority of our method in shape segmentation and 3D semantic correspondence.

## 2   Related Work

**Dense Shape Correspondence**   While there are many dense correspondence works for organic shapes [6, 15, 18, 29, 30, 37, 42, 48], due to space, our review focuses on methods designed for man-

made objects, including optimization and learning-based methods. For the former, most prior works build correspondences only at a *part* level [2,21,25,44,55]. Kim *et al.* [27] propose a diffusion map to compute point-based "fuzzy correspondence" for every shape pair. This is only effective for a small collection of shapes with limited shape variations. [26] and [20] present a template-based deformation method, which can find point-level correspondences after rigid alignment between the template and target shapes. However, these methods only predict coarse and discrete correspondence, leaving the structural or topological discrepancies between matched parts or part ensembles unresolved.

A series of learning-based methods [19,34,49,53,54] are proposed to learn local descriptors, and treat correspondence as 3D semantic landmark estimation. E.g., ShapeUnicode [34] learns a unified embedding for 3D shapes and demonstrates its ability in correspondence among 3D shapes. However, these methods require *ground-truth* pairwise correspondences for training. Recently, Chen *et al.* [8] present an unsupervised method to estimate 3D structure points. Unfortunately, it estimates a *constant* number of *sparse* structured points. As shapes may have diverse part constitutions, it may not be meaningful to establish the correspondence between all of their points. Groueix *et al.* [17] also learn a parametric transformation between two surfaces by leveraging cycle-consistency, and apply to the segmentation problem. However, the deformation-based method always deforms all points on one shape to another, even the points from a non-matching part. In contrast, our *unsupervisedly* learnt model can perform pairwise *dense* correspondence for any two shapes of a man-made object.

**Implicit Shape Representation** Due to the advantages of continuous representation and handling complicated topologies, implicit functions have been adopted for learning representations for 3D shape generation [10,32,33,38], encoding texture [36,43,45], and 4D reconstruction [35]. Meanwhile, some works [23,24] leverage the implicit representation together with a deformation model for shape registration. However, these methods rely on the deformation model, which might prevent their usage for topology-varying objects. Slavcheva *et al.* [46] present an approach which implicitly obtains correspondence for organic shapes by predicting the evolution of the signed distance field. However, as they require a Laplacian operator to be invariant, it is limited to small shape variations. Recently, some extensions have been proposed to learn deep structured [12,13] or segmented implicit functions [9], or separate implicit functions for shape parts [39]. However, instead of at a part level, we extend implicit functions for unsupervised dense shape correspondence.

## 3 Proposed Method

Let us first formulate the dense 3D correspondence problem. Given a collection of 3D shapes of the same object category, one may encode each shape $\mathbf{S} \in \mathbb{R}^{n \times 3}$ in a latent space $\mathbf{z} \in \mathbb{R}^d$. For any point $p \in \mathbf{S}_A$ in the source shape $\mathbf{S}_A$, dense 3D correspondence will find its semantic corresponding point $q \in \mathbf{S}_B$ in the target shape $\mathbf{S}_B$ if a semantic embedding function (SEF) $f : \mathbb{R}^3 \times \mathbb{R}^d \to \mathbb{R}^k$ is able to satisfy

$$\left( \min_{q \in \mathbf{S}_B} ||f(p, \mathbf{z}_A) - f(q, \mathbf{z}_B)||_2 \right) < \tau, \quad \forall p \in \mathbf{S}_A. \tag{1}$$

Here the SEF is responsible for mapping a point from its 3D Euclidean space to the semantic embedding space. When $p$ and $q$ have sufficiently similar locations in the semantic embedding space, they have similar semantic meaning, or functionality, in their respective shapes. Hence $q$ is the corresponding point of $p$. On the other hand, if their distance in the embedding space is too large ($\geq \tau$), there is not a corresponding point in $\mathbf{S}_B$ for $p$. If SEF could be learned for a small $\tau$, the corresponded point $q$ of $p$ can be solved via $q = f^{-1}(f(p, \mathbf{z}_A), \mathbf{z}_B)$, where $f^{-1}(:,:)$ is the inverse function of $f$ that maps a point from the semantic embedding space back to the 3D space. Therefore, the dense correspondence amounts to learning the SEF and its inverse function.

Toward this goal, we propose to leverage the topology-free implicit function, a conventional shape representation, to jointly serve as SEF. By assuming that corresponding points are similar in the embedding space, we explicitly implement an inverse function mapping from the embedding space to the 3D space, so that the learning objectives can be more conveniently defined in the 3D space rather than the embedding space. Both functions are jointly learned with an occupancy loss for accurate shape representation, and a self-reconstruction loss for the inverse function to recover itself. In addition, we propose a cross-reconstruction loss enforcing two objectives. One is that the two functions can deform source shape points to be sufficiently close to the target shape. The other is that corresponding offset vectors, $\overrightarrow{pq}$, are locally smooth within the neighbourhood of $p$.

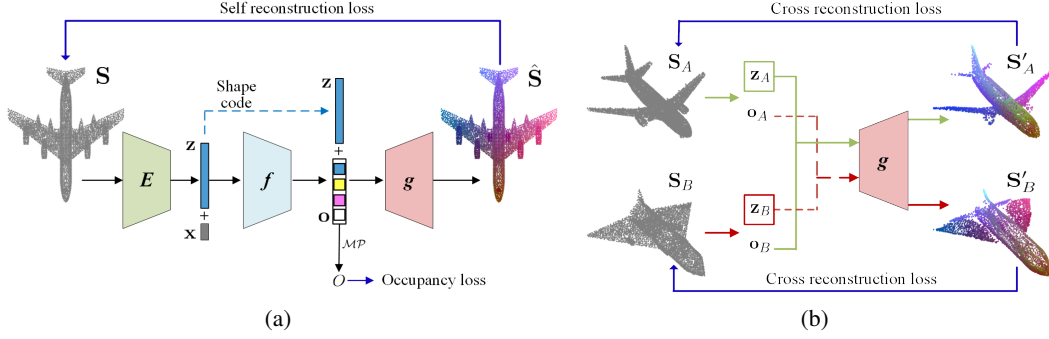

(a)

(b)

Figure 2: **Model Overview.** (a) Given a shape $\mathbf{S}$, PointNet $E$ is used to extract the shape feature code $\mathbf{z}$. Then a part embedding $\mathbf{o}$ is produced via a deep implicit function $f$. We implement dense correspondence through an inverse function mapping from $\mathbf{o}$ to recover the 3D shape $\hat{\mathbf{S}}$. (b) To further make the learned part embedding consistent across all the shapes, we randomly select two shapes $\mathbf{S}_A$ and $\mathbf{S}_B$. By swapping the part embedding vectors, a cross reconstruction loss is used to enforce the inverse function to recover to each other.

## 3.1 Implicit Function and Its Inverse

**Implicit Function** As in [10, 33], a shape $\mathbf{S} \in \mathbb{R}^{n \times 3}$ is first encoded as a shape code $\mathbf{z} \in \mathbb{R}^d$ by a PointNet $E : \mathbb{R}^{n \times 3} \rightarrow \mathbb{R}^d$ [40]. Given the 3D coordinate of a query point $\mathbf{x} \in \mathbb{R}^3$, the implicit function assigns an occupancy probability $O$ between 0 and 1, where 1 indicates $\mathbf{x}$ is inside the shape, and 0 outside. This conventional function can not serve as SEF, given its simple 1D output. Motivated by the unsupervised part segmentation [9], we adopt its branched layer as the final layer of our implicit function, whose output is denoted by $\mathbf{o} \in \mathbb{R}^k$ in Fig. 2: $f : \mathbb{R}^3 \times \mathbb{R}^d \rightarrow \mathbb{R}^k$. A max-pooling operator ($\mathcal{MP}$) leads to the final occupancy $O = \mathcal{MP}(\mathbf{o})$ by selecting one branch, whose index indicates the unsupervisedly estimated part where $\mathbf{x}$ belongs to. Conceptually, each element of $\mathbf{o}$ shall indicate the occupancy value of $\mathbf{x}$ w.r.t. the respective part. Since $\mathbf{o}$ appears to represent the occupancy of $\mathbf{x}$ w.r.t. all semantic parts of the object, the latent space of $\mathbf{o}$ can be the desirable semantic embedding, and thus we term $\mathbf{o}$ as the part embedding vector (PEV) of $\mathbf{x}$. In our implementation, $f$ is composed of 3 fully connected layers each followed by a LeakyReLU, except the final output (Sigmoid).

**Inverse Implicit Function** Given the objective function in Eqn. 1, one may consider that learning SEF, $f$, would be sufficient for dense correspondence. However, this has two issues. 1) To find correspondence of $p$, we need to compute $f^{-1}(f(p, \mathbf{z}_A), \mathbf{z}_B)$, *i.e.*, assuming the output of $f(q, \mathbf{z}_B)$ equals $f(p, \mathbf{z}_A)$ and solve for $q$ via iterative back-propagation. This can be inefficient during inference. 2) It is easier to define shape-related constraints or losses between $f^{-1}(f(p, \mathbf{z}_A), \mathbf{z}_B)$ and $q$ in the 3D space, than those between $f(q, \mathbf{z}_B)$ and $f(p, \mathbf{z}_A)$ in the embedding space. To this end, we define the inverse implicit function to take PEV $\mathbf{o}$ and the shape code $\mathbf{z}$ as inputs, and recover the corresponding 3D location: $g : \mathbb{R}^k \times \mathbb{R}^d \rightarrow \mathbb{R}^3$. We use a multilayer perception (MLP) network to implement $g$. With $g$, we can efficiently compute $g(f(p, \mathbf{z}_A), \mathbf{z}_B)$ via forward passing, without iterative back-propagation.

## 3.2 Training with Loss Functions

We jointly train our implicit function and inverse function by minimizing three losses: occupancy loss $\mathcal{L}^{occ}$, self-reconstruction loss $\mathcal{L}^{SR}$, and cross-reconstruction loss $\mathcal{L}^{CR}$, *i.e.*,

$$\mathcal{L}^{all} = \mathcal{L}^{occ} + \mathcal{L}^{SR} + \mathcal{L}^{CR}, \tag{2}$$

where $\mathcal{L}^{occ}$ measures how accurately $f$ predicts the occupancy of the shapes, $\mathcal{L}^{SR}$ enforces $g$ is an inverse function of $f$, and $\mathcal{L}^{CR}$ strives for part embedding consistency across all shapes in the collection. We first explain how we prepare the training data, then detail our losses.

**Training Samples** Given a collection of $N$ raw 3D surfaces $\{\mathbf{S}_i^{raw}\}_{i=1}^N$ with consistent upright orientation, we first normalize the raw surfaces by uniformly scaling the object such that the diagonal of its tight bounding box has a constant length and make the surfaces watertight by converting them to voxels. Following the sample scheme of [10], for each shape, we obtain $K$ spatial points $\{\mathbf{x}_j\}_{j=1}^K$

and their occupancy label $\{\tilde{O}_j\}_{j=1}^K \in \{0,1\}$, which is 1 for the inside points and 0 otherwise. In addition, we uniformly sample $n$ *surface points* to represent 3D shapes, resulting in $\{\mathbf{S}_i\}_{i=1}^N$.

**Occupancy Loss** This is a $L_2$ error between the label and estimated occupancy of all shapes:

$$\mathcal{L}^{occ} = \sum_{i=1}^N \sum_{j=1}^K \|\mathcal{MP}(f(\mathbf{x}_j, \mathbf{z}_i)) - \tilde{O}_j\|_2^2. \tag{3}$$

**Self-Reconstruction Loss** We supervise the inverse function by recovering input surface points $\mathbf{S}_i$:

$$\mathcal{L}^{SR} = \sum_{i=1}^N \sum_{j=1}^n \|g(f(\mathbf{S}_i^{(j)}, \mathbf{z}_i), \mathbf{z}_i) - \mathbf{S}_i^{(j)}\|_2^2, \tag{4}$$

where $\mathbf{S}_i^{(j)}$ is the $j$-th vertex of shape $\mathbf{S}_i$.

**Cross-Reconstruction Loss** The cross-reconstruction loss is designed to encourage the resultant PEVs to be similar for densely corresponded points from any two shapes. As in Fig. 2, from a shape collection we first randomly select two shapes $\mathbf{S}_A$ and $\mathbf{S}_B$. The implicit function $f$ generates PEVs $\mathbf{o}_A$ ($\mathbf{o}_B$) given $\mathbf{S}_A$ ($\mathbf{S}_B$) and their respective shape codes $\mathbf{z}_A$ ($\mathbf{z}_B$) as inputs. Then we swap their PEVs and send the concatenated vectors to the inverse function $g$: $\mathbf{S}_A'^{(j)} = g(\mathbf{o}_B^{(j)}, \mathbf{z}_A), \mathbf{S}_B'^{(j)} = g(\mathbf{o}_A^{(j)}, \mathbf{z}_B)$. If the part embedding is point-to-point consistent across all shapes, the inverse function should recover each other, *i.e.*, $\mathbf{S}_A' \approx \mathbf{S}_A, \mathbf{S}_B' \approx \mathbf{S}_B$. Towards this goal, we exploit several loss functions to minimize the pairwise difference between those shapes:

$$\mathcal{L}^{CR} = \lambda_1 \mathcal{L}^{CD} + \lambda_2 \mathcal{L}^{EMD} + \lambda_3 \mathcal{L}^{nor} + \lambda_4 \mathcal{L}^{smo}, \tag{5}$$

where $\mathcal{L}^{CD}$ is Chamfer distance (CD) loss, $\mathcal{L}^{EMD}$ Earth Mover distance (EMD) loss, $\mathcal{L}^{nor}$ surface normal loss, $\mathcal{L}^{smo}$ smooth correspondence loss, and $\lambda_i$ are the weights. The first three terms focus on the shape similarity, while the last one encourages the correspondence offsets to be locally smooth.

*Chamfer distance loss* is defined as:

$$\mathcal{L}^{CD} = d_{CD}(\mathbf{S}_A, \mathbf{S}_A') + d_{CD}(\mathbf{S}_B, \mathbf{S}_B'), \tag{6}$$

where CD is calculated as [40]: $d_{CD}(\mathbf{S}, \mathbf{S}') = \sum_{p \in \mathbf{S}} \min_{q \in \mathbf{S}'} \|p - q\|_2^2 + \sum_{q \in \mathbf{S}'} \min_{p \in \mathbf{S}} \|p - q\|_2^2$.

*Earth mover distance loss* is defined as:

$$\mathcal{L}^{EMD} = d_{EMD}(\mathbf{S}_A, \mathbf{S}_A') + d_{EMD}(\mathbf{S}_B, \mathbf{S}_B'), \tag{7}$$

where EMD is the minimum of sum of distances between a point in one set and a point in another set over all possible permutations of correspondences [40]: $d_{EMD}(\mathbf{S}, \mathbf{S}') = \min_{\Phi:\mathbf{S} \to \mathbf{S}'} \sum_{p \in \mathbf{S}} \|p - \Phi(p)\|_2$, where $\Phi$ is a bijective mapping.

*Surface normal loss* An appealing property of implicit representation is that the surface normal can be analytically computed using the spatial derivative $\frac{\partial \mathcal{MP}(f(\mathbf{x}, \mathbf{z}))}{\partial \mathbf{x}}$ via back-propagation through the network. Hence, we are able to define the surface normal distance on the point sets.

$$\mathcal{L}^{nor} = d_{nor}(\mathbf{n}_A, \mathbf{n}_A') + d_{nor}(\mathbf{n}_B, \mathbf{n}_B'), \tag{8}$$

where $\mathbf{n}_*$ is the surface normal of $\mathbf{S}_*$. We measure $d_{nor}$ by the Cosine similarity distance: $d_{nor}(\mathbf{n}, \mathbf{n}') = \frac{1}{n} \sum_i (1 - \mathbf{n}_i \cdot \mathbf{n}_i')$, where $\cdot$ denotes the dot-product.

*Smooth correspondence loss* encourages that the correspondence offset vectors $\Delta \mathbf{S}_{AB} = \mathbf{S}_B' - \mathbf{S}_A$, $\Delta \mathbf{S}_{BA} = \mathbf{S}_A' - \mathbf{S}_B$ of neighboring points are as similar as possible to ensure a smooth deformation:

$$\mathcal{L}^{smo} = \sum_{a,a' \in \mathcal{N}(a)} \|\Delta \mathbf{S}_{AB}^{(a)} - \Delta \mathbf{S}_{AB}^{(a')}\|_2 + \sum_{b,b' \in \mathcal{N}(b)} \|\Delta \mathbf{S}_{BA}^{(b)} - \Delta \mathbf{S}_{BA}^{(b')}\|_2, \tag{9}$$

where $a \in \mathbf{S}_A$, $b \in \mathbf{S}_B$, $\mathcal{N}(a)$ and $\mathcal{N}(b)$ are neighborhoods for $a$ and $b$ respectively.

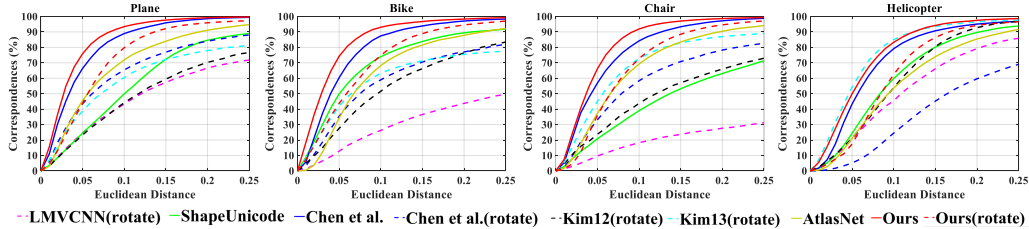

Figure 3: Correspondence accuracy for 4 categories in the BHCP benchmark. The dashed lines indicate the methods are rotation-invariant and for the unaligned setting. All baseline results are quoted from [8, 26].

### 3.3 Inference

During inference our method can offer both shape segmentation and dense correspondence for 3D shapes. As each element of PEV learns a compact representation for one common part of the shape collection, the shape segmentation of $p$ is the index of the element being max-pooled from its PEV.

As both the implicit function $f$ and its inverse $g$ are point-based, the number of input points to $f$ can be arbitrary during inference. Given two point sets $\mathbf{S}_A$, $\mathbf{S}_B$ with shape codes $\mathbf{z}_A$ and $\mathbf{z}_B$, $f$ generates PEVs $\mathbf{o}_A$ and $\mathbf{o}_B$, and $g$ outputs cross-reconstructed shape $\mathbf{S}'_A$. For any query point $p \in \mathbf{S}_A$, a preliminary correspondence may be found by a nearest neighbour search in $\mathbf{S}'_A$: $q' = \arg\min_{q' \in \mathbf{S}'_A} \|p - q'\|_2$. Knowing the index of $q'$ in $\mathbf{S}'_A$, the same index in $\mathbf{S}_B$ refers to the final correspondence $q \in \mathbf{S}_B$. Here, the nearest neighbor search might not be optimal as it limits the solution to the already sampled points in $\mathbf{S}_B$. An alternative is that, once the preliminary correspondence $q'$ is found, within its neighbourhood, we can search an surface point who is closer to $p$ than $q'$. As our input shapes are densely sampled, this alternative does not provide notable benefits, and thus we use the first approach. Finally, we compute the correspondence confidence as $\mathcal{C} = 1 - \mathcal{D}$, where $\mathcal{D} = \|\mathbf{o}_A(i_p, :) - \mathbf{o}_B(i_q, :)\|_2$ is normalized to the range of $[0, 1]$, and $i_p$ is the index of $p$ in $\mathbf{S}_A$. Since the learned part embedding is discriminative among different parts of a shape, the distance of PEVs is suitable to define the confidence. When $C$ is larger than a pre-defined threshold $\tau'$, this $p \rightarrow q$ correspondence is valid; otherwise $p$ has no correspondence.

### 3.4 Implementation Detail

Our method is trained in three stages: 1) PointNet $E$ and implicit function $f$ are trained on sampled point-value pairs via Eqn. 3. 2) $E$, $f$, and inverse function $g$ are jointly trained via Eqn. 3 and 4. 3) We jointly train $E$, $f$ and $g$ with $\mathcal{L}^{all}$. In experiments, we set $n = 8,192$, $d = 256$, $k = 12$, $\tau' = 0.2$, $\lambda_1 = 10$, $\lambda_2 = 1$, $\lambda_3 = 0.01$, $\lambda_4 = 0.1$. We implement our model in Pytorch and use Adam optimizer at a learning rate of 0.0001 in all stages.

## 4 Experiments

### 4.1 3D Semantic Correspondence

**Data** We evaluate on 3D semantic point correspondence, a special case of dense correspondence, with two motivations: 1) no database of man-made objects has ground-truth dense correspondence; 2) there is far less prior work in dense correspondence for man-made objects, than the semantic correspondence task, which has strong baselines for comparison. Thus, to evaluate semantic correspondence, we train on ShapeNet [7] and test on BHCP [26] following the setting of [8, 19]. For training, we use a subset of ShapeNet including plane (500), bike (202), chair (500) categories to train 3 individual models. For testing, BHCP provides ground-truth semantic points (7-13 per shape) of 404 shapes including plane (104), bike (100), chair (100), helicopter (100). We generate all pairs of shapes for testing, *e.g.*, 9,900 pairs for bike. The helicopter is tested with the plane model as [8, 19] did. As BHCP shapes are with rotations, prior works test on either one or both settings: aligned and unaligned (*i.e.*, 0° vs. arbitrary relative pose of two shapes).

**Baseline** We compare our work with multiple state-of-the-art (SOTA) baselines. Kim12 [27] and Kim13 [26] are traditional optimization methods that require part label for templates and employ

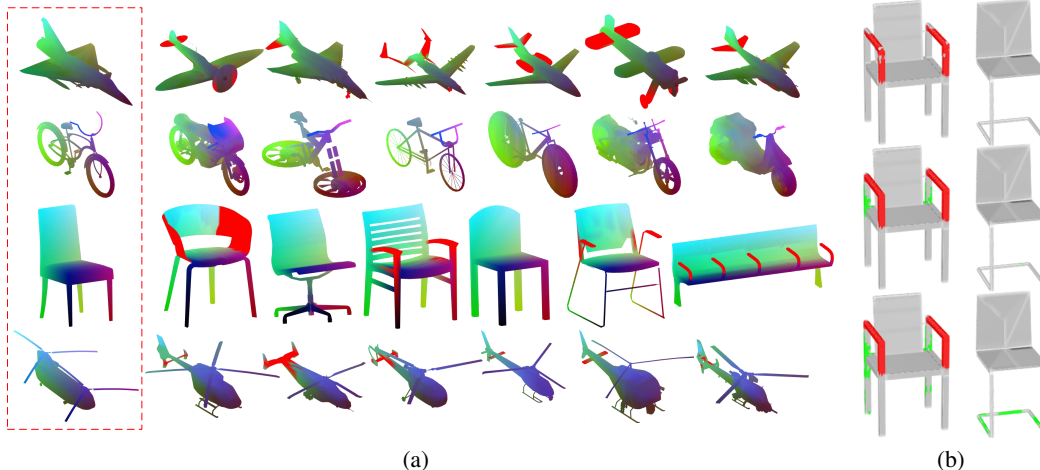

(a)                                                                                                    (b)

Figure 4: (a) Dense correspondences in 4 categories. Each row shows one target shape $\mathbf{S}_B$ (red box) and its pair-wise corresponded 6 source shapes $\mathbf{S}_A$. Given a spatially colored $\mathbf{S}_B$, the $p \rightarrow q$ correspondence enables to assign $p \in \mathbf{S}_A$ with the color of $q \in \mathbf{S}_B$, or with red if $q$ is non-existing. (b) For one pair, the non-existence correspondences are impacted by the confidence threshold ($\tau' = 0.2, 0.5$, and $0.7$ from top to bottom).

collection-wise co-analysis. LMVCNN [19], ShapeUnicode [34], AtlasNet2 [11] and Chen *et al.* [8] are all learning based, where [19, 34] require ground-truth correspondence labels for training. Despite [8] only estimates a fixed number of sparse points, [8] and ours are trained **without** labels. As optimization-based methods and [19] are designed for the unaligned setting, we also train a rotation-invariant version of ours by supervising $E$ to predict an additional rotation matrix and applying it to rotate the input point before feeding to $f$.

**Results** The correspondence accuracy is measured by the fraction of correspondences whose error is below a given threshold of Euclidean distances. As in Fig. 3, the solid lines show the results on the aligned data and dotted lines on the unaligned data. We can clearly observe that our method outperforms baselines in plane, bike and chair categories on aligned data. Note that Kim13 [26] has a slightly higher accuracy than ours on the helicopter category, likely due to the fact that [26] tests with the helicopter-specific model, while we test on the *unseen* helicopter category with a plane-specific model. At the distance threshold of $0.05$, our method improves on average $17\%$ accuracy in 4 categories over [8]. For unaligned data, our method achieves competitive performance as baselines. While it has the best AUC overall, it is worse at the threshold between $[0, 0.05]$. The main reason is the implicit network itself is sensitive to rotation. Note that this comparison shall be viewed in the context that most baselines use extra cues during training or inference, as well as high inference speed of our learning-based approach. Some visual dense correspondence results are shown in Fig. 4(a). Note the amount of non-existent correspondence is impacted by the threshold $\tau'$ as in Fig. 4(b). A larger $\tau'$ discovers more subtle non-existence correspondences. This is expected as the division of semantically corresponded or not can be blurred for some shape parts.

By only finding the closest points on aligned 3D shapes, we report its semantic correspondence accuracy as the black curve in Fig. 6(a). Clearly, our accuracy is much higher than this "lower bound", indicating our method doesn't rely much on the canonical orientation. To further validate on noisy real data, we evaluate on the Chair category with additive noise $\mathcal{N}(0, 0.02^2)$ and compare with Chen *et al.* [8]. As shown in Fig. 6(a), the accuracy is slightly worse than testing on clean data. However, our method still outperforms the baseline on noisy data.

**Detecting Non-Existence of Correspondences** Our method can build dense correspondences for 3D shapes with different topologies, and automatically declare the non-existence of correspondence. The experiment in Fig. 3 cannot fully depict this capability of our algorithm as no semantic point was annotated on a non-matching part. Also, there is no benchmark providing the non-existence label between a shape pair. We thus build a dataset with $1,000$ paired shapes from the chair category of ShapeNet part dataset. Within a pair, one has the arm part while the other does not. For the former, we annotate 5 arm points and 5 non-arm points based on provided part labels. As correspondences don't exist for the arm points, we can utilize this data to measure our detection of non-existence of

Table 1: Unsupervised segmentation on ShapeNet part. We use #parts in evaluation and $k$=12 for all 8 models.

| Shape (#parts) | plane (3) | bag (2) | cap (2) | chair (3) | chair* (4) | mug (2) | skateboard (2) | table (2) | Aver. |
|---|---|---|---|---|---|---|---|---|---|
| Segmented parts | body,tail, wing+engine | body, handle | panel, peak | back+seat, leg, arm | back, seat, leg, arm | body, handle | deck, wheel+bar | top, leg+support | |
| BAE-Net [9] | 80.4 | 82.5 | 87.3 | 86.6 | 83.7 | 93.4 | 88.1 | 87.0 | 86.1 |
| Proposed | 81.0 | 85.4 | 87.9 | 88.2 | 86.2 | 94.7 | 91.6 | 88.3 | **88.0** |

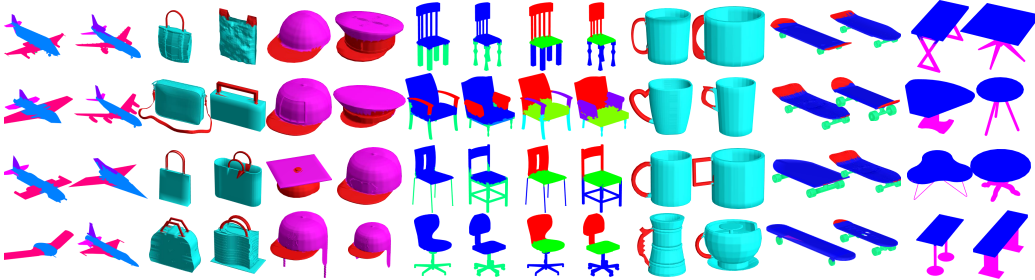

Figure 5: Qualitative results of our unsupervised segmentation in Tab. 1: 8 shapes in each of the 8 categories.

correspondence. Based on our confidence scores, we report the ROC in Fig. 6(b). The 96.28% AUC shows our strong capability in detecting no correspondence.

## 4.2 Unsupervised Shape Segmentation

In testing, unlike prior template-based [26] or feature point estimation methods [8], we don't need to transfer any segmentation labels. Thus, we only compare with the SOTA unsupervised segmentation method BAE-Net [9]. Following the same protocol [9], we train category-specific models and test on the same 8 categories of ShapeNet part dataset [52]: plane $(2, 690)$, bag $(76)$, cap $(76)$, chair $(3, 758)$, mug $(184)$, skateboard $(152)$, table $(5, 271)$, and chair* (a joint chair+table set with $9, 029$ shapes). Intersection over Union (IoU) between prediction and the ground-truth is a common metric for segmentation. Since unsupervised segmentation is not guaranteed to produce the same part counts exactly as the ground-truth, *e.g.*, combining the seat and back of a chair as one part, we report a modified IoU [9] measuring against both parts and part combinations in the ground-truth. As in Tab. 1, our model achieves a consistently higher segmentation accuracy for all categories than BAE-Net. As BAE-Net is very similar to our model trained in Stage 1, these results show that our dense correspondence task helps the PEV to better segment the shapes into parts, thus producing a more semantically meaningful embedding. Some visual results of segmentation are shown in Fig. 5.

## 4.3 Ablations and Visualizations

**Shape Representation Power of Implicit Function** We hope our novel implicit function $f$ still serves as a shape representation while achieving dense correspondence. Hence its shape representation power needs to be evaluated. Following the setting of Tab. 1, we first pass a ground-truth point set from the test set to $E$ and extract the shape code $\mathbf{z}$. By feeding $\mathbf{z}$ and a grid of points to $f$, we can reconstruct the 3D shape by Marching Cubes. We evaluate how well the reconstruction matches the ground-truth point set. The average Chamfer distance (CD) between ours and branched implicit function (BAE-Net) on the 8 categories is $3.5 \pm 1.3$ and $5.4 \pm 2.7$ ($\times 10^3$), respectively. The lower CD shows that our novel design of semantic embedding actually improves the shape representation.

**Loss Terms on Correspondence** Since the point occupancy loss and self-reconstruction loss are essential, we only ablate each term in the cross-reconstruction loss for the chair category. Correspondence results in Fig. 7(a) demonstrate that, while all loss terms contribute to the final performance, $\mathcal{L}^{CD}$ and $\mathcal{L}^{smo}$ are the most crucial ones. $\mathcal{L}^{CD}$ forces $\mathbf{S}'_B$ to resemble $\mathbf{S}_B$. Without $\mathcal{L}^{smo}$, it is possible that $\mathbf{S}'_B$ may resemble $\mathbf{S}_B$ well, but with erroneous correspondences locally.

**Part Embedding over Training Stages** The assumption of learned PEVs being similar for corresponding points motivates our algorithm design. To validate this assumption, we visualize the PEVs of 10 semantic points, defined in Fig. 7(b), with their ground-truth corresponding points across 100 chairs. The t-SNE visualizes the $100 \times 10$ $k$-dim PEVs in a 2D plot with one color per semantic point, after each training stage. The model after Stage 1 training resembles BAE-Net. As in Fig. 7(c),

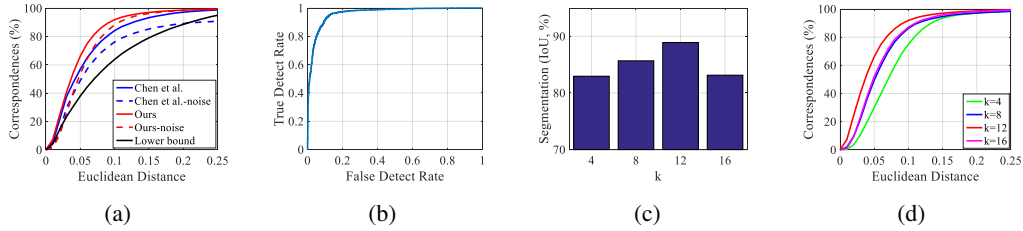

Figure 6: (a) Additional semantic correspondence results for the chair category in BHCP. (b) ROC curve of non-existence of correspondence detection. (c) Shape segmentation and (d) 3D semantic correspondence performances on the chair category over different dimensionalities of PEV.

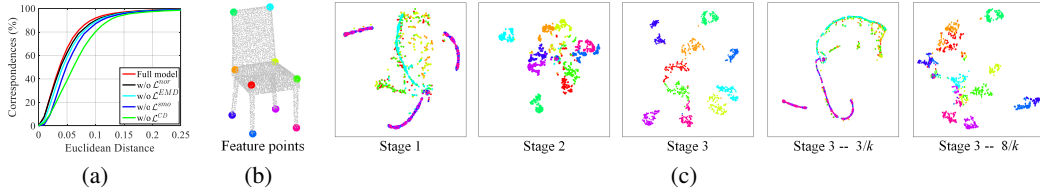

Figure 7: (a) 3D semantic correspondence reflecting the contribution of our loss terms. (b) 10 semantic points overlaid with the shape. (c) The t-SNE of the estimated PEVs over 3 training stages. Points of the same color are the PEVs of ground-truth corresponding points in 100 chairs. 10 colors refer to the 10 points in (b). In $3/k$, only the 3 elements of PEVs max-pooled for 3-part chair segmentation are fed to t-SNE; $8/k$ uses extra 5 elements.

the 100 points corresponding to the same semantic point, *i.e.*, 2D points of the same color, scatter and overlap with other semantic (colored) points. With the inverse function and self-reconstruction loss in Stage 2, the part embedding shows more promising grouping of colored points. Finally, the part embedding after Stage 3 has well clustered and more discriminative grouping, which means points corresponding to the same semantic location do have similar PEVs. The improvement trend of part embedding across 3 stages shows the effectiveness of our loss design and training scheme.

**One-hot vs. Continuous Embedding** Ideally, BAE-Net [9] should output a one-hot vector before $\mathcal{MP}$, which would benefit unsupervised segmentation the most. In contrast, our PEVs prefer a continuous embedding rather than one-hot. To better understand PEV, we compute the statistics of Cosine Similarity (CS) between the PEVs and their corresponding one-hot vectors: $0.972 \pm 0.020$ (BAE-Net) vs. $0.966 \pm 0.040$ (ours). This shows our learnt PEVs are *approximately* one-hot vectors. Compared to BAE-Net, our smaller CS and larger variance are likely due to the limited network capability, as well as our encouragement to learn a continuous embedding benefiting correspondence.

**Dimensionality of PEV** Fig. 6(c) and 6(d) show the shape segmentation and semantic correspondence results over the dimensionality of PEV. Our algorithm performs the best in both when $k = 12$. Despite unsupervisedly segmenting chairs into 3 parts, the extra $9 = (k - 3)$ dimensions of PEV benefit the finer-grained task of correspondence (Fig. 7(c)), which in turns help segmentation.

**Computation Time** Our training on one category (500 samples) takes $\sim$8 hours to converge with a GTX1080Ti GPU, where 1, 1, and 6 hours are spent at Stage 1, 2, 3 respectively. In inference, the average runtime to pair two shapes ($n$=8,192) is $0.21$ second including runtimes of $E$, $f$, $g$ networks, neighbour search and confidence calculation.

## 5  Conclusion

In this work, we propose a novel framework including an implicit function and its inverse for dense 3D shape correspondences of topology-varying objects. Based on the learnt semantic part embedding via our implicit function, dense correspondence is established via the inverse function mapping from the part embedding to the corresponding 3D point. In addition, our algorithm can automatically calculate a confidence score measuring the probability of correspondence, which is desirable for man-made objects with large topological variations. The comprehensive experimental results show the superiority of the proposed method in unsupervised shape correspondence and segmentation.

## Broader Impact

Product design (*e.g., furniture* ) is labor extensive and requires expertise in computer graphics. With the increasing number and diversity of 3D CAD models in online repositories, there is a growing need for leverage them to facilitate future product development due to their similarities in function and shape. Towards this goal, our proposed method provide a novel unsupervised paradigm to establish dense correspondence for topology-varying objects, which is a prerequisite for shape analysis and synthesis. Furthermore, as our approach is designed for generic objects, its application space can be extremely wide.

## Acknowledgement

The authors would like to thank the reviewers and area chairs for their valuable comments and suggestions. We acknowledge Vladimir G. Kim and Nenglun Chen for sharing data and results.

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
