[Supplementary Material]

# Learning Implicit Functions for Topology-Varying Dense 3D Shape Correspondence — Supplementary Material

**Feng Liu**     **Xiaoming Liu**
Department of Computer Science and Engineering
Michigan State University, East Lansing MI 48824
{liufeng6, liuxm}@msu.edu

In this supplementary material, we provide:

$\diamond$ Implementation details, including network structures and training details.

$\diamond$ Additional experimental results, including expressiveness of the inverse implicit function and visualization of the correspondence confidence score.

## 1   Implementation Details

### 1.1   Network Structures

**PointNet Encoder $E$.**   To extract the shape code, we adopt PointNet [3] like architecture as our encoder. The detailed architecture of $E$ is depicted in Fig. 1(a). The Encoder takes a point set as input and generates a 256-dim shape latent code $\mathbf{z}$.

**Implicit Function $f$.**   The implicit function network follows the work of [1] (unsupervised case). The implicit function takes the shape code $\mathbf{z}$ and a spatial point $(x, y, z)$ as inputs and predicts the part embedding vector (PEV) $\mathbf{o}$. As shown in Fig. 1(b), it is composed of 3 fully connected (FC) layers each of which is applied with $Leaky\ ReLU$, except the final output is applied a $Sigmoid$ activation.

**Inverse Implicit Function $g$.**   The inverse implicit function is also implemented as an MLP, which is composed of 8 FC layers each of which is applied with $Leaky\ ReLU$, except the final output is applied a $Tanh$ activation. As shown in Fig 1(c), the inverse implicit function network inputs the PEVs and shape latent code, and recover the corresponding 3D points.

### 1.2   Training Details

**Sampling Point-Value Pairs.**   The training of implicit function network needs point-value pairs. Following the sampling strategy of [2], we obtain the paired data $\{\mathbf{x}_j, \tilde{O}_j\}_{j=1}^{K}$ offline. $\mathbf{x}_j, \tilde{O}_j$ are the spatial point and the corresponding occupancy label. We sample points from the voxel models in different resolutions: $16^3$ ($K = 4,096$), $32^3$ ($K = 8,192$) and $64^3$ ($K = 32,768$) in order to train the implicit function progressively.

**Training Process**   We summarize the training process in Tab. 1. In Stage 1, we adopt a progressive training technique [2] to train our implicit function on gradually increasing resolution data ($16^3 \rightarrow 32^3 \rightarrow 64^3$), which stabilizes and significantly speeds up the training process.

| (a) PointNet-based encoder | (b) Implicit function | (c) Inverse implicit function |

Figure 1: **Network Architectures.** (a) The PointNet-based encoder network. A shape code $\mathbf{z} \in \mathbb{R}^{256}$ is predicted from the input point set. $\mathcal{MP}$ denotes the max-pooling operator. (b) The implicit function network is composed of 3 fully connected layers, denotes as "FC". The shape code is concatenated, denoted as "+", with the xyz query, making a 259-dim vector, and is provided as input to the first layer. The $Leaky\ ReLU$ activation is applied to the first 2 FC layers while the part embedding vector $\mathbf{o}$ is obtained with a $Sigmoid$ activation. Finally, a max-pooling operator gives the final occupancy value $O$. (c) The inverse implicit function network is also implemented as a MLP, which is composed of 8 FC layers. Specifically, it takes PEV $\mathbf{o}$ and the shape code $\mathbf{z}$ as inputs, and recover the corresponding 3D location.

Table 1: Stages of the training process.

|  | Network | Loss |
|---|---|---|
| Stage 1 | $E, f$ | $\mathcal{L}^{occ}$ |
| Stage 2 | $E, f, g$ | $\mathcal{L}^{occ}$ and $\mathcal{L}^{SR}$ |
| Stage 3 | $E, f, g$ | $\mathcal{L}^{all}$ |

## 2 Additional Experimental Results

A supplementary video is provided to visualize additional results, explained as follows.

### 2.1 Expressiveness of Inverse Implicit Function

Given our inverse implicit function, we are able to cross-reconstruct each other between two paired shapes by swapping their part embedding vectors. Further, we can interpolate shapes both in shape latent space and 3D space and maintain the point-level correspondence consistently.

**Cross-Reconstruction Performance.** We first show the cross-reconstruction performances in the supplementary video. From a shape collection, we can randomly select two shapes $\mathbf{S}_A$ and $\mathbf{S}_B$. Their shape codes $\mathbf{z}_A$ and $\mathbf{z}_B$ can be predicted by the PointNet encoder. With their respectively generated PEVs $\mathbf{o}^A$ and $\mathbf{o}^B$, we can swap their PEVs and send the concatenated vectors to the inverse function and obtain $\mathbf{S}'_A = g(\mathbf{o}_B, \mathbf{z}_A), \mathbf{S}'_B = g(\mathbf{o}_A, \mathbf{z}_B)$. As shown in the video, the cross reconstructions closely resemble each other, even with different part constitutions. Here, we also provide the cross-reconstruction performance of two additional object categories: car and table.

**Interpolation in Latent Space.** An alternative way to explore the correspondence ability of the inverse implicit function, is to evaluate the interpolation capability of the inverse implicit function. In this experiment, we first interpolate shapes in the latent space $\tilde{\mathbf{z}} = \alpha\mathbf{z}_A + (1-\alpha)\mathbf{z}_B$ ($\alpha \in [0, 1]$), and send the concatenated vectors ($\tilde{\mathbf{z}}$ and $\mathbf{o}_A$) to the inverse function. As observed in the video, our inverse implicit function generalizes well the different shape deformations. Moreover, the correspondences are point-to-point consistent across all the deformations. It also demonstrates that the learned part embedding is discriminative among different parts of shape and point-wise consistent among different shapes.

**Latent Interpolation Comparison.** We compare the latent interpolate capability with conventional implicit function. For the conventional implicit function, we sample a grid of points and pass them to the implicit function to obtain its value. With the threshold of $0.5$, we obtain the surface points. As can be observed in the video, the interpolation performance of our inverse implicit function is

better than conventional implicit function in shape generation and deformation. Furthermore, our interpolations are point-to-point correspondence across all the deformations.

**Interpolation in 3D Space.** We also show the interpolation capability of the corresponding points in the 3D space in the video. Given the estimated dense correspondence, we can compute the correspondence offset vectors $\Delta \mathbf{S}_{AB} = \mathbf{S}'_B - \mathbf{S}_A$ for all corresponding pairs of points. Assuming we interpolate the correspondence in $N$ video frames, for each frame we move all points of $\mathbf{S}_A$ by the amount of $\frac{1}{N}\Delta \mathbf{S}_{AB}$ and show the moved points. It can be observed that our deformed shape is meaningful and a semantic blending of two shapes. In addition, the correspondence offsets are locally smooth in the 3D space.

### 2.2 Visualization of the Correspondence Confidence Score

To further visualize the correspondence confidence score, we provide the confidence score maps for some examples in Figure 4 of the paper. As shown in the video, the confidence score can show the probability around corresponded points between the target shape (red box) and its pair-wise source shapes. For example, for the source shapes with arms, we can clearly see the confidence scores of the arm part is significantly lower than other parts.