[Reviews · NeurIPS 2020]

Review 1

Summary and Contributions: The paper presents an approach to estimate correspondence across shapes of different topologies via learned implicit functions -- or more precisely learned fields. The backbone is very similar to an unsupervised segmentation network [8]. A shape point is encoded into k-dimensional vector. The occupancy is obtained by max pooling the k-dim vector, and the discovered part is simply the index of the max. The key contribution and idea however is a cross-reconstruction loss. Given two shapes A and B, the key constraint is to point-wise reconstruct B from A. Every point in A is encoded into a k-dim vector. Based on the k-dim vector and a global shape code z_B, a corresponding point in B is reconstructed. This involved two mappings f and g which are jointly learned.

Strengths: The idea is quite nice and I while it resembles ideas for disentanglement, it has not been explored for unsupervised discovery of correspondences across shapes. There are several ablation studies. The text is mostly clear (although the notation and math is unprecise at times)

Weaknesses: - Training this method is probably involved and expensive. For example, the chamfer requires reconstructing the full shape S{A,B}^\prime and then finding closest points during training. Since implicit functions are trained point-wise, I am not sure how it is implemented in practice. There is no mention of computational time to train the model. - The results on ShapeNet look nice. I wonder how this would look like for more complex objects, like humans. I would have been much more convinced if there would be a sanity check testing the method on established benchmarks like Faust[4] and DFaust Bogo et al. CVPR'17 where ground truth correspondence is available. Sure those have the same topology but it would validate the approach. - I am doubtful about how useful the approach is. The shapes are scaled and perfectly aligned. It would be good to see the correspondence accuracy of just finding the closest points after aligning the data. Also the unsupervised segmentation network does not require the contributions mentioned here and the performance is already quite good 86.1 vs 88.0. It would be nice to see segmentation results for the baseline as well. ** Post Rebuttal ** Thanks for the rebuttal, it was very helpful. The results on FAUST are appreciated as it shows that the method can work on more complex shapes such as humans and also that the method is robust (to an extent) to noisy scans (FAUST scans are noisy with holes etc.). The authors addressed my main concerns and the results presented in the rebuttal look good. Hence, I'm updating my score.

Correctness: yes.

Clarity: The text is clear. The math notation is imprecise sometimes. The authors know what they mean, but what is written does not match I think. For example: - The defined inverse f^-1 does not even take as input the output domain of f() R^k, so technically it can not be called the inverse. Also g() is defined later anyway, why not use g from the start? - There is confusion between \mathbf{o} and \mathbf{O} and O. Not sure what each is, but the text uses different versions. Fig 2 uses \mathbf{O} although it is never defined. - Line 164: now g outputs S_A' which is a full pointset? g is defined as a pointwise decoded mapping to R^3. It is clear from context what authors mean, but still confusing. - Notation S_i(j,:): I assume that is the j-th point of shape S_i. not sure what (,:) notation is doing. Not defined either. - Line 278: I believe what is meant here is forces S_B^\prime to be close to S_B (instead of S_B). There are a few other errors like this in the text

Relation to Prior Work: The work is well positioned and the related work is nicely covered. There are related recent works (CVPR'20, ECCV'20) that could be included. - Bhatnagar et al. Combining Implicit Function Learning and Parametric Models for 3D Human Reconstruction. ECCV'20: predicts part correspondences to a template via implicit functions, just like this work. But the approach is supervised. - Line 90: Deng et al. Neural Articulated Shape Approximation. ECCV'20 recently learns deformable implicit function. - Suwajanakorn et al. Discovery of Latent 3D Keypoints via End-to-end Geometric Reasoning. NIPS'18. Very relevant. Unsupervised discovery of keypoints. - Jakab et al. Self-supervised Learning of Interpretable Keypoints from Unlabelled Videos. Keypoint detection. - Atzmon et al. Controlling Neural Level Sets. NIPS'19 and other recent papers from Yaron Lipman's group. - The paper uses a point encoding based on a global latent code and a point. The work of Chibane et al. Implicit Functions in Feature Space for 3D Shape Reconstruction and Completion CVPR'20 extracts multi-scale features at continuous locations. Given that the task is fine grained correspondences, it is possible this kind of encoding would be more localized and less global, leading to better correspondences. Note that there are many others that find keypoints across classes of objects, and animals but those are in 2D. Still they could be cited.

Reproducibility: Yes

Additional Feedback: Reproducible because code will be released. I think training this model is not straightforward. Main suggestions to improve (the current manuscript or follow-up work) 1) Demonstrate results on FAUST and DFAUST against ground truth correspondence. 2) Fix math notation and errors. 3) Add recent related works. 4) Consider using more local encodings of shape as in Chibane et al. Implicit Functions in Feature Space for 3D Shape Reconstruction and Completion. CVPR'20. Otherwise I am afraid, the network relies too much on the canonical orientation of shapes, and reasons less about shape structure to find correspondence. Overall, this is a solid paper and I like it. I do not rate higher because I have concerns on how practical it is to train the model, and how well it works against ground truth dense correspondences (FAUST, DFAUST). I can be persuaded to give a higher score if concerns are addressed.


Review 2

Summary and Contributions: This paper proposes an unsupervised method for 3D shape dense correspondence. Building on the recently introduces neural implicit functions, the method can handle changes in topology

Strengths: + the paper is well written and easy to follow. + analysis and experiments are adequate + The method introduces several interesting concepts i did not see before, including: part embedding that's also used as point embedding; inverse function that maps back from semantic space to 3D space; and the interesting CR loss. + The invertable mapping between 3D and semantic space is very interesting

Weaknesses: - What strikes me as surprising is the direct usage of the function f as the part embedding and point embedding. This needs further justifications, as ideally a BAE-like network should assign a hard per-part occupancy score of {0,1}. e.g. if the point coordinate belongs to the wing of an airplane, then the ideal f should output a one-hot vector. Clearly networks struggle in doing so, thus creating non binary outputs. However, this is more of a bug than a feature. Yet, the authors here exploit this approximation by making the network encode per-point subtleties into this approximated 1-hot. My intuition tells me this is not a principled way of doing it and perhaps a point embedding should be an intermediate prediction from which another few layers convert it into a part embedding? - The CR loss assumes all sampled points on the source have a corresponding point on the target. However, as explained by the authors, often part may appear and/or disappear. How is this being addressed? minor: line 109: a more detailed definition of f^-1 is recommended. In particular, it should be made explicit that f^-1 takes as input both the semantic embedding as well as the shape code and retrieves the coordinate. references: Probably relevant is this recent work that also uses implicit neural function to register between a partial and a full shape: "The Whole Is Greater Than the Sum of Its Nonrigid Parts"

Correctness: See above

Clarity: Yes. See above

Relation to Prior Work: See above

Reproducibility: Yes

Additional Feedback: I'm happy with the rebuttal and appreciate the FAUST experiment. I also appreciate the attempt to shed more light onto the PEV vs. one-hot however I still think a more principled approach might be suitable here. Other than that i'm happy with the high score i gave and will keep it as my final evaluation.


Review 3

Summary and Contributions: The paper proposes a method for part segmentation and dense correspondences estimation that is trained on categories of 3D point clouds of “man-made objects” in an unsupervised way. The paper builds on BAE-Net [8]: it learns an autoencoder, where the latent code consists of the shape embedding and the implicit function. The latter, in turn, can produce for a 3D point x, a vector of part probabilities and occupancy probability (the latter is supervised during training). These embeddings are supervised to be decodable back to the original 3D point cloud. If the input is not rigidly aligned, the camera viewpoint supervision is needed. Finally (but actually the novel part), the paper uses the swap (cross-reconstruction) loss to disentangle the shape code from the implicit occupancy function. Evaluating those problems is difficult, but the paper makes a good effort, and shows SOTA results on both parts segmentation and correspondences, in addition to an ablation study.

Strengths: Novelty. The paper addresses a relatively new but important problem with no established task formalisation and evaluation protocol, and makes methodological improvements. In particular, allowing the prediction of no-correspondence for a point is new and practical. Turning the discrete parts segmentation model into the continuous dense correspondence one by the means of the swap loss is a new and interesting idea. The paper is relevant to the part of the community who work on 3D computer vision. The paper is generally well written. The hypotheses are supported by experiments. In particular, in Section 4.2 the authors created a new dataset annotation by labelling arms of the chairs to measure how well the method can predict lack of correspondence for a point. The ablation study is sufficient.

Weaknesses: The paper evaluates only on the synthetic data, so it is unclear if the results will hold for the noisy real data (alas, this seems to be a standard practice in the subfield). Given the difficulty to obtain annotations, showing visual results on real scans would already be a big step forward. The paper claims to train in an unsupervised way, however it requires that the objects are aligned (or otherwise camera viewpoints should be known to be able to learn them). This is a big limitation in practice, so the paper can be more direct about it rather then just mention in the experiments section. Knowing the correct object pose already gives a good first approximation for canonical coordinates, see e.g. NOCS paper (Wang, 2019). There are some concerns on correctness and clarity, see below. UPD. The authors addressed all my concerns, except for maybe EMD loss assuming 1:1 correspondences, where the answer is not convincing. I recommend accepting the paper. Wang, H., Sridhar, S., Huang, J., Valentin, J., Song, S., & Guibas, L. J. (2019). Normalized Object Coordinate Space for Category-Level 6D Object Pose and Size Estimation. IEEE Conference on Computer Vision and Pattern Recognition, 2637–2646.

Correctness: The components of the swap loss, Chamfer distance and EMD, are similar, but EMD assumes 1:1 correspondences between the points. Is it correct to assume it, given that the paper explicitly aims to model the missing correspondences like the chair with missing arms? The paper evaluates dense correspondences on the annotated sparse semantic correspondences, which is fine for the lack of better alternative, however the paper may be straightforward about the biases it creates. Semantic correspondences are annotated at more salient points (e.g. ends of chair legs), which are easier to detect, so the error on the uniformly sampled points would probably be higher.

Clarity: The paper is generally clear. Some minor issues: * line 198 says that the L2 distance between embeddings is normalised to be in [0, 1]; how specifically is it normalised? * in lines 258 and on, how specifically is IOU modified? The results show that some parts can be degenerate (in some cases, only two parts are used). Does the new metric encourage predicting fewer parts?

Relation to Prior Work: The review is good; the main concern here is distinguishing the contribution from BAE-Net, but I think the paper also does a good job in that too.

Reproducibility: Yes

Additional Feedback: Typos / wording: * line 13: if there is ← if there is one, * line 18: statistical modeling // do you mean more specifically active shape models? * line 20: one can correspond an arbitrary point ← one can establish correspondence between an arbitrary point OR one can match an arbitrary point; * line 59: objects with topology-Varying variations ← categories with diverse topology; * line 119: to be sufficiently closer ← to be sufficiently close; * line 155 and on: inside points // this reads like the points inside the watertight hull (occupancy == 1), while I think the points on the surface are implied; * line 159: to encourage that the resultant PEVs are indeed similar ← to encourage the resultant PEVs to be similar; * line 238: efficient inference speed ← high inference speed / fast inference.


Review 4

Summary and Contributions: Existing dense correspondence algorithms have problems with various topology and deep implicit function can not produce dense correspondence. This work addresses the problem by proposing an implicit dense correspondence network. Specifically, the network maps a 3D point along with a latent shape code to a part embedding space, where the location represents the part identity. Then an inverse implicit module is trained to recover the 3D point given the latent shape code and the PEV. To make the training self-supervised, the authors propose self reconstruction supervision and cross reconstruction supervision. Experiments show superior performance of the proposed approach and a valid application in shape segmentation is shown to demonstrate its potential application and influence on other topics.

Strengths: 1. The proposed method addresses two problems together: the traditional dense correspondence algorithm has problem with topology variation; implicit shape representation is robust to topology variation but can not provide dense correspondence. The work will be appreciated in topics such as surface registration, statistical shape modeling, and shape segmentation. 2. The idea of using self reconstruction loss and cross reconstruction loss is novel and effective for unsupervised learning of dense correspondence. 3. State of the art performance according to the evaluation and comparison.

Weaknesses: 1. It would be better to compare with AtlasNet, which is not an implicit representation but can deal with various topology and produces dense correspondence in a certain sense. 2. It could be better if the authors can show whether this dense correspondence estimation affects the shape reconstruction quality.

Correctness: The claims are fair and backed up by the results. Methods and evaluation are valid.

Clarity: Yes, the paper is well written. It's easy to follow and mathematical formulations are correct. Minor comment, I think 'self-supervised' is more appropriate than 'unsupervised'. The proposed training uses self reconstruction error and cross reconstruction error that are supervised by the involved single shape or shape pair itself. But it is also known that in many literatures when people say 'unsupervised' they refer to 'self-supervised'.

Relation to Prior Work: Yes, relations and differences with dense correspondence algorithms and deep implicit methods are clearly addressed.

Reproducibility: Yes

Additional Feedback:

[Author Response · NeurIPS 2020]



Figure R. 1: (a) Geodesic error comparison of ours (unsup.), FMNet [28] (sup.) and Halimi *et al.* [16] (unsup.) on FAUST. (b) One target shape and 4 pair-wise corresponded source shapes. (c) Additional semantic correspondence results for chair category in BHCP.

We thank the reviewers for their comments. All reviewers are positive about our *novelty* and *compelling results*. We
address reviewer's concerns as follows, and will revise the paper accordingly, including missing citations, typos, etc.

**Review1Q1. Computation time.** Implemented as a custom C++/CUDA extension, our Chamfer distance calculation
is efficient, *e.g.*, $\sim 42.8ms$ for two sets of point clouds $[B{\times}N{\times}3]$, where the batch size $B{=}10$, the number of vertex
$N{=}4,096$. Although the implicit function is point-wise, the $K$ query 3D locations can be constructed as a 3D tensor
$[B{\times}K{\times}3]$ and passed to the fully connected layers. Our training on one category (500 samples) takes $\sim$6 hours (1, 1,
and 4 hours for Stage 1, 2, 3 respectively) to converge with a GTX1080Ti GPU. In inference, the average runtime to pair
two shapes ($N{=}4,096$) is $22.3ms$ including runtime of $E$, $f$, $g$ networks, neighbour search and confidence calculation.

**R1Q2. Dense correspondence on FAUST dataset.** As suggested, we conduct additional experiments on FAUST
humans dataset, with two SOTA baselines: supervised (FMNet [28]) and unsupervised (Halimi *et al.* [16]) methods. We
follow the experimental protocol of [16, 28], and set $N{=}8,192$, $d{=}256$. Note that we use **real scans** in both training
and testing. The ground-truth densely aligned shapes are used for evaluation. As shown in Fig. R.1(a), our method, as a
unsupervised method, outperforms the unsupervised method [16]. Fig. R.1(b) visualizes the estimated correspondences.

**R1Q3. Aligned vs. unaligned data. *i)*** By only finding the closest points on aligned 3D shapes, we report its semantic
correspondence accuracy as the black curve in Fig. R.1(c), following the setting of Sec. 4.1. Clearly, our accuracy is
much higher than this 'lower bound', indicating our method doesn't rely much on the canonical orientation. ***ii)*** In fact,
for practical applicability, we did train a rotation-invariant model for unaligned test setting (L226-228 and Fig. 5 in the
main paper). Our method achieves competitive performance as optimization-based baselines.

**R1Q4. Segmentation comparison.** Unlike ours, the correspondence baselines cannot automatically predict segmen-
tation. They require additional templates to transfer segmentation labels. Thus we only compared with the SOTA
unsupervised segmentation method BAE-Net. Here, we report segmentation results (IoU: $78.5\%$) of Chen *et al.* [7] on
the chair category by using a single template of 3 parts: back+seat, leg, arm, which is worse than ours ($88.9\%$).

**Review2Q1. Justifications on $f$ and part embedding vector (PEV). *i)*** On the test set of chair category in segmen-
tation experiment, we measure the statistics of Cosine Similarity between the PEVs and their corresponding one-hot
vectors: $0.972 \pm 0.200$ (BAE-Net), $0.966 \pm 0.401$ (ours). This shows PEVs are continuous and *approximately* one-hot
vectors. We view this approximation not a bug, but a by-product of limited network capability, and leverage it to learn
PEV. ***ii)*** Per your suggestion, we retrain a model by using the 384-dim feature (1 layer before **o** as Fig. 1(b) in **Supp.**)
as point embedding. As shown in Fig. R.1(c), the correspondence accuracy (purple curve) is worse than ours. ***iii)*** Based
on these results and t-SNE plots of PEVs in Fig. 7, we believe the PEVs of surface points can serve as point embedding.

**R2Q2. CR loss for missing part.** The CR loss is the key to ensure that corresponding points in two shapes share the
*same* PEV (Eqn. 1). When defining CR loss for two shapes with a mis-match part, there are two strategies: 1) pursue
correspondences for all points first, even on mis-match part; and detect mis-match part via the PEV-based confidences
$\mathcal{C}$. 2) define CR loss "only" for highly-confident corresponded points while ignoring points on mis-match part. Strategy
2 requires confidence calculation, which is impossible when PEV has not yet satisfied Eqn. 1. We chose Strategy 1, as it
optimizes PEV to satisfy Eqn. 1, during which points on mis-match part are likely outliers, and reflected in lower $\mathcal{C}$.

**Review3Q1. Evaluation on real data. *i)*** To validate on noisy real data, we evaluate on the BHCP (normalized) testing
data with additive noise $\mathcal{N}(0, 0.02^2)$ and compare with Chen *et al.* [7] and AtlasNet [14]. As shown in Fig. R.1(c), the
accuracy is slightly worse than testing on clean data. However, our method still outperforms baselines on noisy data. ***ii)***
In R1Q2, we use real scans in FAUST dataset for testing, which also illuminates our effectiveness on real data.

**R3Q2. Train on aligned data.** We will clarify the training data requirements early in Sec. 1. Please also refer to R1Q3.

**R3Q3. EMD and biases issues. *i)*** Please refer to R2Q2. ***ii)*** We agree that there are biases our evaluation due to biased
annotation (only on salient points). However, our unsupervised framework encourages dense corresponding of *all*
points, *without* using salient point supervision. Hence, our model itself has no semantic correspondence bias.

**Review4Q1. Compare with AtlasNet on BHCP.** Following the same experimental setting, we report the semantic
correspondence accuracy of AtlasNet [14] in Fig. R.1(c). Our method outperforms AtlasNet.

**R4Q2. Impact on shape representation.** We did show our improved shape representation power in Sec. 4.4 (L267).

[Meta-Review · NeurIPS 2020]

This submission proposes an unsupervised approach to learning correspondences between 3D shapes of different topologies. It initially received four reviews with all positive scores (7,8,7,9). The reviewers appreciated novelty and originality, as well as quality of empirical evaluation and presentation. The rebuttal addressed some of the remaining concerns, which resulted in an increase of scores to (8,8,7,9). For these reasons, the AC's recommendation is to accept this submission for oral presentation.